# Development and validation of an automated machine for self-injury assessment via young Koreans' natural writings

**Seoyoung Kim**[1], **Dong-gwi Lee**[2]*

1 Yonsei University Psychological Science Innovation Research Center, Yonsei University, Seoul, Republic of Korea, 2 Department of Psychology, Yonsei University, Seoul, Republic of Korea

* lee82@yonsei.ac.kr

**Data Availability Statement:** A part of the K-SITR's codebook is presented in a Supporting Information table, and the Python codes are available at a public repository Github (https://

## Abstract

Self-injury is common in all countries, and 20% of South Korean youths experience self-injury. One of the barriers to assessment and treatment planning is the tendency of young self-injurers to conceal their identities. Following a new stream of research that uses online text data to assess psychological symptoms as they are described in online posts, this study developed a computerized machine that can analyze South Korean self-injurers' writing in assessing their self-injury severity. Based on 16,645 online posts, Study 1 developed a machine called the Korean Self-Injurious Text Reviewer (K-SITR) using Latent Dirichlet Allocation topic modeling and machine learning. The K-SITR's text-assessment results were statistically indistinguishable from those of professional counselors. Study 2 confirmed the validity of the K-SITR through a survey of 47 young Koreans who had experienced self-injury. Results showed that the K-SITR scores converged with participants' self-injury frequency and duration and discriminated from other heterogenous factors. The K-SITR also had incremental validity over two popular self-injury questionnaires. This study provides a new measure that may reduce the tendency of young self-injurers to self-conceal compared to traditional direct-item questionnaires.

## Introduction

Self-injury refers to deliberate harm to one's body, *with or without* suicidal intent outside socially sanctioned activities [1]. A meta-analysis by Swannell et al. [2] showed that self-injury was most prevalent among youth (17.2% of adolescents and 13.4% of young adults). Various reasons for self-injury have been identified, including regulation of negative emotions, self-punishment, and avoidance of responsibility [3, 4].

In addition, Voss et al. [5] found that nearly 40% of adolescents and young adults ($N = 1,180$) with a history of self-injury endorsed suicidal behavior. Suicide survivors tend to report multiple episodes of self-injury [6]. Reviews of studies (e.g., [7]) suggest that the suicidal intent of self-injurers varies from time to time in terms of certainty or degree. Repeated self-injury carries a risk that the young self-injurers may adapt to lethal methods over time due to

github.com/kimalexis1129/KSITR/blob/main/S2.%
20K-SITR%20python%20codes%20example.py).

**Funding:** The author(s) received no specific funding for this work.

**Competing interests:** The authors have declared that no competing interests exist.

habituation/desensitization to physical pain and fear of death [8, 9]. As a result, untreated self-injury can pose detrimental risks to the lives of adolescents, including the possibility of both planned and unplanned (i.e., accidental) suicide [10].

In South Korea, nearly 20% of youth reported experiencing self-injury [11, 12] Other empirical findings (e.g., 17% in China [13]; 17.2% in Vietnam [14], 20.1% in Taiwan [15]) suggest that self-injury among youths may be more prevalent when considering Asian study records. South Korean psychologists proposed that a number of sociocultural characteristics, such as interdependent self-construal, emphasis on adherence to collectivist social norms, high academic pressure and perfectionism, may play a role in the severity of the situation [11, 12].

Given the high prevalence and life-threatening adversities, calls for studies of the young self-injurers have increased in all countries [16]; however, recruiting the young self-injurers into research is one of the greatest barriers [17, 18]. Youths who self-injure often hide their identities because of concerns about social stigma and rejection [19]. This self-concealing tendency prevents youths from self-disclosing and seeking help [6, 19, 20]. The private nature of self-injury hinders recovery and research.

## Efforts to assess self-injury and related work

The self-concealing tendency continues during the treatment process. Whether or not young self-injurers seek professional counseling voluntary (e.g., through school referrals [21]), assessing the severity of clients' cases has been a challenge [22]. Measures of self-injury severity are available; however, Faura-Garcia et al. [23] systematically compared and found that only 17.11% of the 152 published measures could be considered reliable. In addition, clients' tendency to obfuscate and shame due to concerns about social stigma are at odds with survey items that ask directly about self-injury experiences, such as "Have you ever intentionally cut your wrists, arms, or other area(s) of your body?" from the Deliberate Self-Harm Inventory: SHI [24]. Garisch et al. [25] noted that "self-report measures may not be completed honestly or may be seen as invasive by some clients" (p. 100).

Assessing the severity of a self-injury case is critical, especially in the early stages of treatment for making important decisions such as the need for hospitalization and whether to refer the case to another counselor with more expertise [10]. However, the first persons to detect self-injury, such as the young self-injurers' classmates or teachers, may lack sufficient knowledge about the characteristics of young self-injurers and assessment strategies [26]. Particularly in schools where most cases of self-injury are found, school officials including the school counselors reported feeling incompetent in terms of their expertise and counseling resources to assess the severity of cases and make the important decisions, such as whether and where to refer the cases of self-injuring students [21].

## Methodology and proposed work

New attempts to address the assessment difficulty have emerged outside of conventional psychology. For example, the Conference and Labs of the Evaluation Forum (CLEF) has begun running "eRisk" competitions to detect signs of mental health crises online [27, 28]. Beginning in 2017, by examining depression in Internet posts on Reddit [29], recent eRisk assignments asked participating programmers to differentiate between texts implying depression, anorexia, and self-injury [28]. Participating in the eRisk in 2020, Martínez-Castaño et al. [30] successfully submitted a text classifier that can detect early signs of self-injury among users of Reddit platform, based on a collection of previous posts from 42,839 Reddit's self-harm community (subreddit) and an equal number of random posts unrelated to self-injury. To improve the accuracy of the classification between the self-injuring and non-self-injuring users, the

authors, who are computer and information scientists, manually tagged the posts of the "real self-harmers" ([30] p. 5) to use their textual features as a criterion. For an extended eRisk task presented a year later in 2021, Campillo-Ageitos et al. [31] proposed a feature-driven classifier. Based on a relatively small number of Reddit posts ($N = 763$), they used classic text features, such as the number of question marks and exclamation points used in posts, and special text features, such as direct descriptions of self-injury (e.g., methods/instruments, reasons), to predict whether a new post belonged to a self-injuring user.

Various text analytic methods, such as topic modeling and sentiment analysis have been used in conjunction with machine learning techniques to complete the eRisk tasks [27–29]. Similar studies have been presented [30, 31], showing that online text data can be a useful up-to-date source of knowledge when access to the population of interest is limited for personal and/or societal reasons, as in the case of young self-injurers with concerns about social rejection and shame-inducing judgements.

Many Internet posts about self-injury are autobiographical. With proper ethical controls, such as avoiding the collection of personal information, the study of these posts has been recognized for its high ecological validity, as shown by submissions to the eRisk competitions [30–32]. Compared to these computer programmers and information scientists, psychologists tend to remain inactive in the academic use of online textual data [32, 33].

In South Korea, rare but some exploratory results have recently been introduced. Kim et al. [34] text-analyzed 1,242 comments on Internet news articles about self-injuring youths to understand how South Koreans in general construe and respond to the self-injury issue. Lee et al. [35] compared the textual characteristics of a set of online posts about self-injury experiences that had received empathic response comments with a counterpart set that had received indifferent, more cynical comments to infer the reasons for social rejection. These studies have shown that despite the conservative social atmosphere in South Korea [34, 35], self-injury disclosures are common and easily accessible online, as they are on Western online sites. However, despite the high prevalence and ease of finding self-injury-related posts in Korean online communities, few practical studies such as the eRisk, have been attempted in the field of South Korean psychology (e.g., [34]).

## The present study

To make use of the self-injury-related online posts and help counseling practitioners assess the cases of young self-injurers more freely of the clients' self-concealing tendency, this study developed and validated an automated machine, using text-mining and machine-learning techniques with the traditional statistical testing methods of psychology. The machine was designed to examine natural writings of the young self-injurers and provide immediate suggestions about the case (i.e., case severity and expected treatment outcome), just as a counseling professional would do at the intake or in the early stages of counseling.

Compared to self-report surveys of self-injury (e.g., SHI [24]), which directly ask clients about the frequency and methods of self-injury, the assessment of free-written texts can help lower the clients' emotional guard and capture detailed personal characteristics outside the range of survey responses often collected on Likert-type scales [36]. In addition, the machine presented in this study has a methodological advantage over other machines developed by computer and information scientists (e.g., [30, 31]). The previous machines were not reviewed by professional psychologists during the process of their development. They may be technically effective; however, the techniques have rarely been accepted or used in actual counseling practice due to a lack of communication and interactive review process between the computer experts and the psychologists. As a multidisciplinary study, this study sought the opinions of

counseling experts on various cases of self-injury (presented in self-injury-related online posts) and the validity of machine-generated text examination results.

The results of this study not only add new knowledge about the characteristics of self-injuring youths, but also provide a practical tool for practice. The machine's text assessment results are based on a cross-sectional writing; therefore, they are preliminary and may be valid only as a first impression of a case. Nevertheless, the results may help counselors who are struggling with a lack of counseling resources such as novice or school counselors, to make timely decisions, such as making a referral to the hospital or another counselor with more expertise or beginning to plan an intervention.

This study is divided into machine development (Study 1) and validation (Study 2), because of differences in terms of data types (text and survey responses) and analysis methods (text mining, machine learning, and traditional psychological statistics). To the best of our knowledge, this study is the first attempt to use online text data from self-injuring South Korean youth for psychological research beyond exploratory purposes.

## Study 1

In Study 1, a text assessment machine was developed. From a corpus of 16,645 publicly available anonymous Internet posts on self-injury experiences in a popular Korean online community, Naver Knowledge-In (https://kin.naver.com/), 178 posts rated as suitable for this study's objectives by counseling experts were used to train the machine.

The development of the beta version involved surveys of counseling experts, text analyses, and Python computer coding of a hierarchical structure for assessments. The machine was finalized using a beta test. Revisions were made to make the machine's inferential results statistically indistinguishable from judgments by counseling experts in terms of the written cases' levels of case severity and expected (negative) treatment outcomes.

## Materials and methods

### Development of a beta version

**Data collection.** The raw corpus (post $N$ = 16,645) was web-crawled from the Knowledge-In online community using the search query "self-injury" ("자해" in Korean) from January 2020 to July 2021. The Python Selenium module (https://github.com/baijum/selenium-python) and Google Chrome web driver were used to navigate the Internet. Inclusion criteria were applied to select and use high-quality posts with sufficient information on young writers' experiences, as shown in Fig 1.

The screened posts ($N$ = 250) were presented to four supervisor-level counseling experts with an average of 12.75 years ($SD$ = 2.99) of professional counseling career experience (three

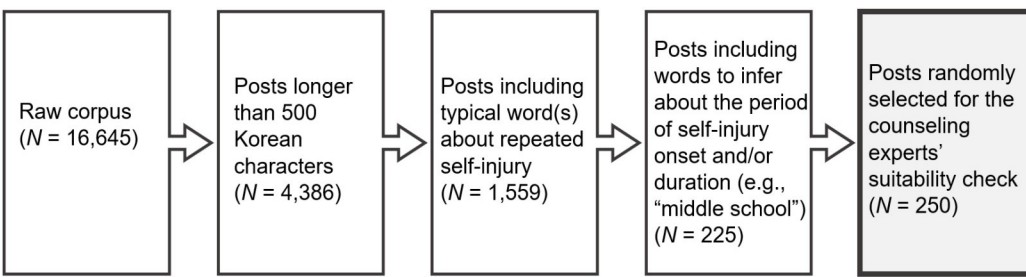

**Fig 1. Data inclusion criteria.**

females and one male, aged 38–52). Each expert rated 125 posts regarding their suitability to this study's objectives on a Likert-type scale from 1 to 7 (1 = "completely unsuitable"; 7 = "completely suitable"). Specifically, the experts were asked to judge the extent to which a post looked suitable for gaining therapeutically useful information on the self-injurious young writer if the person was in intake or early counseling as a new client of the experts. Two experts rated the same posts and showed fair inter-rater consistency (Cohen's kappa = .40) according to the standard suggested by Landis and Koch [37]. Based on the results, 178 posts (hereafter, "training data") that received an average suitability score of 5 or higher were used to train the machine's beta version.

**Training data classification.** To acquire professional judgments on the training data, 18 counselors using diverse approaches (e.g., psychoanalysis, client-centered therapy, and gestalt) participated in a survey (17 females and one male, 33–56 years old, career in counseling $M = 17.67$ years, $SD = 4.51$). Each counselor was shown 18–20 training data posts and asked how they would evaluate the cases if the writers were their clients. For example, an unedited training data post was as follows:

> I started self-injuring from the 2nd grade of middle school. At that time, I was under a lot of stress because I was having a hard time making friends. During gym class, I put my head down and did not move at all for more than 30 minutes. I experienced anxiety symptoms like scratching myself or biting my nails during team projects. When I did this, I felt extremely anxious and by the time I realized it, my arms and nails were red and swollen. I didn't tell my parents. (Leave out the rest)

Two counselors read each post and judged the levels of case severity, expected difficulty in intervention, and expected treatment outcomes on a Likert-type scale ranging from 1 to 7 (1 = relatively not severe/not difficult/not concerned; 7 = very severe/difficult/concerned). Consistencies (Cohen's kappa) were fair between the two raters (.30 for the case severity, .25 for the expected difficulty to intervene, and .24 for the expected treatment outcome).

For more practical information, counselors were also asked to choose rankings from 1 to 5 (ranked 1st = suggested most often) among the intervention priorities (client's traits and personality, thoughts, emotions, past environment, and current environment) for each case. The intervention domains were chosen based on a review of the literature on counseling processes and outcomes [10, 38].

A preliminary check of counselors' responses showed that their judgments of case severity and expected difficulty in intervening largely overlapped (Pearson's $r = .80$); therefore, the two assessments for a given post were merged into one indicator of case severity. Using this application, the training data were classified by level, as shown in Table 1.

**Table 1. Training data sub-sets.**

| Sub-set | Counselors' judgement (n) | Case severity M (SD) | Treatment outcome M (SD) |
|---|---|---|---|
| A | Serious but positive outcome expected (26) | 4.44 (0.38) | 2.88 (0.26) |
| B | Serious and negative outcome concerning (62) | 4.53 (0.40) | 3.94 (0.37) |
| C | Not as serious and positive outcome expected (64) | 3.30 (0.52) | 2.55 (0.52) |
| D | Not as serious but negative outcome concerning (26) | 3.58 (0.37) | 3.65 (0.27) |

Treatment outcome = expected (negative) treatment outcome.

**Text analyses.** Each training dataset was subjected to a series of textual analyses. First, the original posts in a subset were pre-processed by tokenizing its paragraphs into unigram words (hereafter, "tokens") using the KOMORAN Korean morphological analyzer (https://github.com/shineware/KOMORAN). Subsets were left with meaningful tokens by excluding empty morphemes such as pre- and postpositions. After preprocessing, the training data contained 18,486 meaningful tokens ($M$ = 103.85, $SD$ = 61.38). The current training data preserved more tokens than the tokens preserved from other social media posts consisting of a few short sentences (e.g., Twitter/X, see an example in Zhang et al. [39]).

Second, Latent Dirichlet Allocation (LDA) topic modeling [40, 41] was conducted to derive a topic model for each subset. The topic model is a holistic summary of the data, through which focal information can be shown. Topic models include the main causes of self-injury (triggering events) and frequently expressed thoughts and emotions related to self-injury. Text-analytical methods such as topic modeling assume that there are a few essential topics in a collection of writings, although the essentials are not directly observable.

Based on Bayesian inference and Gibbs sampling, LDA extracts essential topics from observable tokens through thousands of random token sampling iterations. Topic modeling analyzes qualitative data such as natural writing in a quantitative way based on token frequencies; therefore, the topic model can provide topic proportion estimates, which indicate the importance of a topic in explaining the subset compared to other topics (see Blei et al. [41] for more on LDA). There are other topic modeling methods suitable for short-text social data exist, such as random projection [42], latent semantic analysis [43], and non-negative matrix factorization [44]; however, this study decided to use the LDA because it has been proposed as one of the most reliable methods for producing topic models with high levels of precision and topic coherence [45].

An efficient topic model has small between- and large within-topic similarities [40, 41]. For instance, subset B, "serious and negative outcome concerning" could have topics about for example "unstable home environment," and "mood issues." Each topic consists of a list of key tokens. Signs of abuse and neglect like "alone" or "bruise" could be found in the "unstable home environment" topic, while more expressions of emotion such as "depressed" or "tears" could be found in the "mood issues" topic.

Based on the LDA results, a codebook was composed to compare and match its elements to the input during the text-assessment procedure (see the translated page in S1 Table). The codebook includes the corresponding subset index (A to D), key tokens, token standardized representative scores based on their frequency, topic numeric indexes (1 to 10), topic titles, and topic proportion estimates.

**Development of the beta version.** A beta version of the machine learning algorithm was developed using hierarchical "if statements," as shown in Fig 2. The lower hierarchy H1, aimed to match an input (presumably a new client's writing about his or her self-injury experience) to the most similar subset, and to draw on the counselors' Likert-type judgments (about case severity and expected treatment outcomes) that had been made for the subset. The first assessment procedure involved screening the tokens in the input and finding a match among the tokens listed in a codebook. If a token of the input matches a token in subset A, the machine calculates the probability score of the subset A by adding the token's representativeness score, as pre-assigned in the codebook. After screening, each subset's probability score indicates the likelihood that the input could be considered similar to the subset's condition. For example, if the subset A received the highest probability score for an input, it meant that the input's condition was likely a case of a seriously depressed client, but a positive outcome was expected.

The second assessment procedure proceeded within the subset designated as the input for the results of the first assessment (H1 in Fig 2). The higher hierarchy (H2 in Fig 2) aimed to

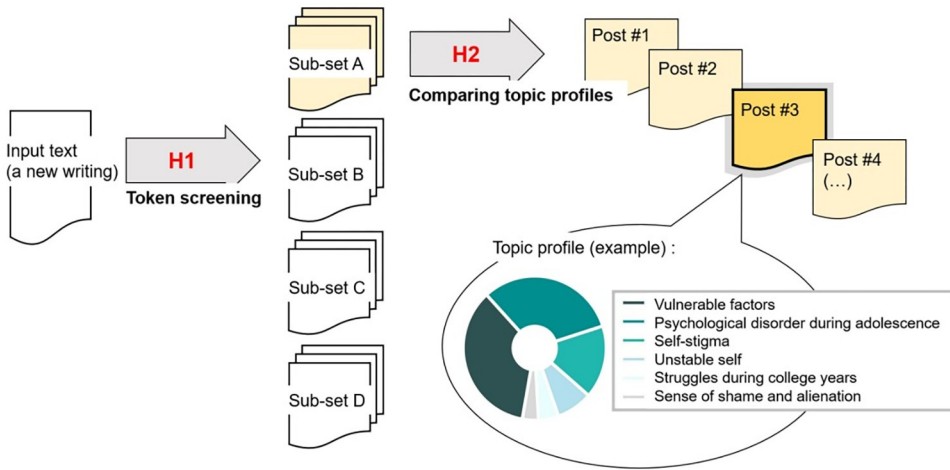

*H1 = lower hierarchy, H2 = higher hierarchy.

**Fig 2. An assessment hierarchy of the machine.** H1 = lower hierarchy, H2 = higher hierarchy.

find a training data post/case belonged to the subset that is specifically similar to the input and retrieve the counselors' ranking-type judgment on the intervention priorities for the case. To achieve this, the machine compared the topical characteristics of the input with those of each of the training data cases within the subset. For an extended example, if an input was found to be most similar to subset A (severe but positive outcome expected), which had a topic model consisting of a number of depressive symptoms such as indicators of maladjustment severe, but less so than topics found from subset B (severe and negative outcome concerning), the input would include some topics from subset A but not all of them due to the input's limited length (one or two paragraphs long) and individual differences. It is likely that the topic profile of the input would resemble a part of the topic model of subset A.

Therefore, to find a training data case specifically similar to the input, the machine converted the input tokens into a Python *dictionary* format consisting of string keys and integer values. The keys were the topic titles and the values were the topic proportion estimates. If the input had no matching tokens for certain topics in the subset, the value of the key remained zero. This dictionary represents the topic profile of the input. For example, if an input was found to have a topic profile summarized as "family conflict, 36; bullying, 0; feelings of inadequacy, 8; interest in treatment, 0", this dictionary implied that the client's depression was closely related to an unstable family environment that had led to negative emotions such as inadequacy. The client may be showing a passive attitude toward the counseling intervention, as the tokens associated with the interest in treatment topic were not present in the client's writing. Unlike some other clients with similar levels of depression severity and outcome expectancy, the client did not show signs of being a victim of bullying. The topic profiles of the input and training data cases were compared until the machine found the best possible match for the input with the most similarities to draw the counselors' ranking-type judgment previously made on the training data case.

### Finalization of the machine

**Beta test.** The beta version, which was structured using two assessment procedures (Fig 2), was tested to evaluate its performance. When there are right and wrong answers, a general beta test can be performed by the researchers who developed the machine; however, the

present machine aimed to emulate counselors' judgments on latent variables, such as self-injury severity. Thus, the beta test involved 30 new posts in a corpus that earned the average suitability score 4 ("moderate") but had not been used as training data and four new counselors (three females and one male, 32–52 years old, average 20 years in professional counseling). All 30 posts were independently shown to the counselors. Counselors were asked to make Likert- and ranking-type judgments as in the previous survey. The Intraclass Correlation Coefficient (ICC) between the four counselors was statistically significant: ICC (2, 4) = .861 ($p <$ .001) for case severity and .767 ($p <$ .001) for expected treatment outcomes. According to Shrout and Fleiss's [46] standards, the ICC shows that counselors are generally consistent when making therapeutic judgments.

At the same time, the posts were assessed using the beta version to compare their results with the counselors' responses using one-way multivariate analysis of variance (MANOVA). The machine aimed to yield statistically indistinguishable results from the counselors.

**Machine revision and finalization.** The beta test showed that the beta version's and human counselors' judgments on the case severity were sufficiently similar [$F$ (1, 28) = 0.34, $p >$ .05]. In contrast, a statistically significant difference was found in the expected (negative) treatment outcome [$F$ (1, 28) = 16.73, $p <$ .05]. On average, the machine ($M$ = 3.49) judged a 0.64 point higher on negative treatment outcomes than the counselors ($M$ = 2.85); therefore, the beta version required revision to enhance its performance.

The revision process involved the following three steps: During the first assessment procedure (H1 in Fig 2), the probability score was specified using two parameters: the relative representativeness score and the absolute number of matching tokens across subsets. Two parameters were used simultaneously to compare the likelihood that an input would be deemed most similar to one of the subsets. Second, a list of tokens that could imply a high risk was selected based on a literature review, as advised by Victor and Klonsky [47] in a meta-analysis of risk factors. Specifically, suicide-related tokens, tokens about practices of cutting (e.g., "cutter knife"), and signs of depression, trauma, and (borderline) personality disorder were given additional points to give stronger weights to the tokens. Finally, the first three tokens that matched during the initial screening of an input were assigned an additional point to their representativeness scores as weights, considering the traits of natural language. In most natural narratives, the main points/arguments tend to precede elaborate examples and evidence [48]. A series of MANOVA were conducted by applying each measure in steps, and no statistical difference was found between the machine and counselors. All analyses were performed using the SPSS 26 statistics software.

The machine was finalized, and the PyFPDF file designer library (pyfpdf.readthedocs.io) was used to add a function that can automatically organize.pdf format client reports downloadable to future users' PCs to maximize the utility of the machine for counseling practices.

## Results of Study 1

Study 1 developed a machine that can assess Korean youths' writing to provide tentative but instant judgments on the levels of the self-injury case's severity and expected treatment outcomes, along with practical suggestions worthy of consideration for counseling cases.

Table 2 presents the 30 tokens most frequently used by young writers in their descriptions of self-injury to illustrate the general characteristics of the online text data used in this study.

The training data were classified into four subsets, and the LDA topic models derived from these subsets are shown in Table 3. Across the subsets, counselors chose clients' emotions as the highest priority for intervention. A subset's topic model can explain the types of risks, triggering events, thoughts, and emotions most relevant to self-injury cases. For instance, subset B

**Table 2. Thirty-most frequently used words by young Korean self-injurers.**

| Rank | Token (freq.) | Rank | Token (freq.) | Rank | Token (freq.) |
|------|---------------|------|---------------|------|---------------|
| 1 | Thoughts (591) | 11 | Study (204) | 21 | Story (110) |
| 2 | Mother (537) | 12 | Reason (202) | 22 | Stress (107) |
| 3 | Friend (422) | 13 | Grade (199) | 23 | Mental (107) |
| 4 | Parents (356) | 14 | Severity (194) | 24 | Problem (106) |
| 5 | People (323) | 15 | Depressed mood (174) | 25 | High school (105) |
| 6 | Depression (307) | 16 | Now (167) | 26 | Elementary school (103) |
| 7 | Self-injury (303) | 17 | Family (135) | 27 | Suicide (102) |
| 8 | School (251) | 18 | Start (132) | 28 | Words (100) |
| 9 | Father (249) | 19 | Counseling (112) | 29 | Academic institute (100) |
| 10 | Middle school (240) | 20 | Mind (110) | 30 | Nowadays (98) |

Freq. = frequency

included references to psychopathologies such as depression, bipolar disorder, and panic disorder. Other subsets included issues related to low self-esteem and maladjustment but relatively fewer clinical symptoms. Based on these differences, the machine compared the probabilities that an input would belong to the subsets to identify the most similar one and proceeded by comparing the input's topic profile with that of the corresponding subset's training data cases.

During the revision of the beta version, subsequent MANOVA results (Table 4) showed that all weights (either based on a token's indication of high risk or the order of its appearance) were necessary to make the machine's inferential results statistically indistinguishable from the counselors' judgments, $F(1, 28) = 3.8$, $p > .05$. The revisions were applied, and the final version was named the Korean Self-Injurious Text Reviewer (K-SITR). Part of the K-SITR Python codes can be found at https://github.com/kimalexis1129/KSITR/blob/main/supp.KSITR_example%20code.py.

## Study 2

Study 2 aimed to validate the K-SITR by comparing it with conventional self-injury measures. It involved a survey of young Koreans who had experienced self-injury but had stopped for over a year to avoid accidentally encouraging self-injury due to exposure to the survey items. The responses were used to check the convergent, discriminant, and incremental validity of the K-SITR. In addition, a counseling expert reviewed the K-SITR client reports (see a translated example in S1 File), provided practical comments, and suggested revisions for refinement. All procedures involving human subjects were approved by the Yonsei University Institutional Review Board (approval number: 7001988-202110-HR-1363-01) on December 6, 2021.

## Materials and methods

### Participants

The participants were 50 young adults (68% female, age $M = 23$, $SD = 1.21$) who had experienced self-injury, which started mostly during adolescence at around the age of 15 (age ranged 7–22). They continued self-injuring for a median of 3 months ($M = 10.57$, $SD = 15.64$) but at least a year and two months had passed since stopping self-injury (stopped for three years and ten months in median). The participants used to self-injure twice a week ($M = 2.87$, $SD = 2.2$),

**Table 3. LDA topic models of the training data sub-sets.**

| Set | Index | Topic | Est. | Topic's key tokens (example) |
|---|---|---|---|---|
| A | 3 | Adolescence psychological issues and delinquency | 39 | Depression, stress, self-esteem |
|  | 8 | Lack of familial support | 19 | Parents, guardian, letter of apology |
|  | 9 | Female struggles | 7 | Female student, victim, bullying |
|  | 1 | Adolescence school maladjustment 1 | 7 | School qualification exam, a juvenile delinquent, atmosphere |
|  | 6 | Familial conflicts 1 | 6 | Stepmother, grandmother, grandfather |
|  | 2 | School violence | 6 | Mental hospital, kicking, a punk |
|  | 4 | Familial conflicts 2 | 5 | Fault finding, cellphone, calves |
|  | 0 | Self-criticism | 4 | Self-hate, garbage, a retard |
|  | 5 | Adolescence school maladjustment 2 | 3 | Break time, schoolyard, a punk |
|  | 7 | Adolescence school maladjustment 3 | 3 | Office of education, Monday, a mess |
| B | 5 | Adolescence psychopathology | 35 | Depression, bipolar disorder, panic disorder |
|  | 6 | Self-stigma | 18 | Garbage, self-hate, self-esteem |
|  | 4 | Self-injury provoking vulnerability factors | 13 | Puberty, bullying, Covid-19 |
|  | 3 | Unstable self | 9 | Stress, confidence, Hikicomori1 |
|  | 1 | Sexual trauma | 7 | Sexual harassment, boyfriend, social phobia |
|  | 2 | Early adulthood psychological issues | 5 | College, sleeping pills, lethargic feeling |
|  | 8 | Sense of shame and alienation | 4 | Self-hate, inferiority, dark age |
|  | 7 | Lack of familial care | 4 | Grandmother, younger brother, smartphone |
|  | 0 | Lack of familial support | 2 | Pressure, bystander, counseling agency |
|  | 9 | Familial conflicts | 2 | Stress, house chores, family affairs |
| C | 5 | Academic stress | 25 | A teacher, Covid-19, school life |
|  | 8 | Familial conflicts—academic grades | 14 | Parents, school report card, guilt |
|  | 2 | Familial conflicts—adolescence career 1 | 14 | Pride, the middle class, middle to low grade |
|  | 1 | Interpersonal issues | 14 | Anxiety disorder, interpersonal relationship, regular days |
|  | 6 | Familial conflicts—parent-child, siblings | 8 | Cellphone, younger brother, stormy period |
|  | 3 | Familial conflicts—adolescence career 2 | 8 | Private high school, middle school for arts, scolding |
|  | 9 | Alienation at home | 6 | Communication, sense of alienation, house chores |
|  | 7 | Peer relationship issues | 6 | Atmosphere, dormitory, an offender |
|  | 4 | Interest in psychological treatment | 4 | Psychiatry, an expert, thought pattern |
|  | 0 | Past domestic violence | 2 | Kindergarten, statue of limitations, child abuse |
| D | 4 | Adolescence mood issues | 32 | Depression, lie, puberty |
|  | 3 | Difficulties in communicating at home | 24 | Parents, stress, communication |
|  | 7 | Lack of surrounding support | 9 | A teacher, family affairs, social phobia |
|  | 2 | Lifestyle issues | 7 | Game addict, Covid-19, garbage |
|  | 8 | Lack of familial care | 5 | Scolding, tuition, cellphone |
|  | 0 | Dysthymia | 5 | Self-criticism, instability, melancholy |
|  | 5 | Sense of alienation | 5 | Psycho, insomnia, self-criticism |
|  | 1 | Interest in psychological treatment | 5 | Psychological test, medical service fee, guardian |
|  | 9 | Peer relationship maladjustment | 5 | Fieldtrip, dormitory, born and bred |
|  | 6 | Sense of inadequacy | 3 | Scolding, a troublesome person, cold sweat |

Est = topic's proportion estimate.

and the most severe participant reported self-injuring ten times a week. The participant recruitment started on December 13 and ended on December 22, 2021, online via Entrust survey panel group (http://entrustsurvey.com/sampling-company-entrust-survey.html). Informed consent was obtained online with participants' electronic signatures.

**Table 4. MANOVA results of the K-SITR beta version's revision.**

| Criterion | Versions | Mean Difference (I-J) | Probability of significance | 95% confidence interval | |
|---|---|---|---|---|---|
| | | | | Lower | Upper |
| Human counselors | Beta before revisions | -0.636* | .003 | -1.116 | -.156 |
| | Weights to risks | -0.636* | .003 | -1.116 | -.156 |
| | Weights to order | -0.481* | .049 | -.961 | -.001 |
| | All weights applied | -0.296 | .452 | -.776 | .184 |

* $p < .05$.

The counseling expert who provided suggestions for the K-SITR topic titles was a 53-year-old female with a specialty in Gestalt and Acceptance Commitment Therapy and a career in professional counseling for 25 years.

## Measures

First, participants were asked to write 500 or more characters (in Korean) about their experiences of self-injury. The participants were reminded of the confidentiality of any information that could be used to identify them to encourage them to open up; however, there were no guidelines or restrictions on what they could write about. Unlike traditional self-injury questionnaires, the K-SITR does not directly ask participants to disclose their methods or frequencies of self-injury.

Subsequently, two popular self-injury questionnaires were included in the survey for comparison with the K-SITR. For ethical and safety reasons, the participants were those who had stopped self-injury at the time of the study; therefore, their responses to the questionnaires did not reflect self-injury within a year. Instead, participants responded retrospectively based on their experiences at the time of self-harm. In addition, two measures for participants' current mood issues (difficulties in regulating emotions and symptoms of mood disorders) and a measure related to similar yet distinguishable behaviors from self-injury (externalized aggression) were used. Participants' self-injury duration (from onset to the last episode), average weekly frequency, and period of cessation were asked independent of the measures.

**Functional Assessment of Self-Mutilation: FASM.**   FASM is a 41-item measure developed by Lloyd et al. [49] and validated in Korea by Kwon and Kwon [50]. This study used the first 11 items on the frequency of self-injury per method and other severity-related information such as hospitalization experience. Using a 7-point Likert-type scale (0 = "not at all," 6 = "more than 6 items"), a higher total score of the 11 items indicates more severe self-injury, as suggested by Lloyd et al. [49] and applied in Kwon and Kwon [50]. The internal consistency (Cronbach's a) of the items was .80 in Kwon and Kwon [50] at validation and .84 in this study. The other part of the FASM assesses the functions of self-injury, such as punishing oneself or avoiding social responsibilities, which were not of interest in this study.

**Self-Harm Inventory: SHI.**   The SHI is a 20-item scale developed by Sansone et al. [51] to assess the severity of self-injury. For improved readability, this study used a version validated and revised by Lee and Lee [52]. The SHI asks for self-harm frequency per method, including indirect self-destructive behaviors (e.g., torture with overly self-critical thoughts and engaging in unsafe sex). Using a 4-point Likert-type scale (1 = "hardly," 4 = "always"), a higher total score indicates more severe self-harm. Internal consistency was .76 in Lee [53] upon revision and .85 in this study.

**Difficulties in Emotion Regulation Scale: DERS.**   The DERS is a 36-item measure to assess difficulties in emotion regulation developed by Gratz and Roemer [54] and validated in

Korean by Choi [55]. This study used 18 items in three factors: having trouble with impulse-control (five, items), lack of awareness (seven items), and limited access to emotion regulation strategy (six items), using a 5-point Likert scale (1 = "hardly," 5 = "always"). A higher score indicates more difficulty. The internal consistencies of the three factors were .90, .83, and .81, respectively, in Choi's [55] study upon validation, and .94, .94, and .88, respectively, in this study.

**Brief Symptoms Inventory-18: BSI-18.** The BSI-18 is an 18-item measure used to assess the symptoms of mood disorders, developed by Derogatis [56] and validated in Korea by Park et al. [57]. BSI-18 asks about symptoms of depression, anxiety, and somatization using a 5-point Likert scale (1 = "never," 5 = "very severe"). Higher scores indicate more severe mood issues. The Global Severity Index (GSI) has a cutoff score of 63 for screening clinical patients. The internal consistency of the three factors was .80, .81, and .73, respectively, in Park et al. [57] upon validation, and .89, .91, and .87, respectively, in this study.

**Aggression Questionnaire: AQ.** The AQ is a 29-item measure to assess aggression, developed by Buss and Perry [58] and validated in Korean by Seo and Kwon [59]. This study used nine items for the *physical aggression* factor which is a representative form of externalization of negative emotion using a 5-point Likert scale (1 = "never," 5 = "always"). Higher scores indicated a higher likelihood of externalizing negative emotions toward other people. Internal consistency of the factor was .85 in Seo and Kwon [59] upon validation and .84 in this study.

## Analyses

As the participants provided written information about their (past) self-injury experiences and responses to the survey, a correlation analysis was conducted between the three measures (K-SITR, FASM, and SHI) of self-injury severity, directly reported self-injury information (duration and frequency), mood issues (BSI-18), and externalization (AQ) to check the K-SITR's convergent and discriminant validity.

The incremental validity of the K-SITR over the FASM and SHI was also tested. Before running the regression analyses, statistical assumptions were checked, and it was found that the participants' self-injury duration as a dependent variable was not suitable due to its violation of normality and homogeneity of variance assumptions. Hierarchical multiple regression analyses were conducted to check whether the K-SITR could explain a significant amount of the variance in the participants' self-injury frequency, in addition to that explained by the FASM and SHI. All analyses were performed using the SPSS 26 statistics software.

## Results of Study 2

Study 2 aimed to verify the validity of K-SITR. The participants were deemed psychologically stable at the time of the survey, as their GSI ($M$ = 18.32, $SD$ = 14.85) was far below the clinical cutoff of 63, and their level of difficulty in regulating emotions ($M$ = 43.74, $SD$ = 12.26) was much lower than the level found in a clinical sample ($M$ = 89.33, $SD$ = 22.64) in Hallion et al. [60]. Among the 50 writings, the K-SITR succeeded in drawing results from 47 cases. The remaining three were omitted, because the number of tokens found in these cases was insufficient to make reliable inferences.

As shown in Table 5, Pearson's *r* correlations between the K-SITR and self-injury information directly asked participants (duration and frequency) showed that the K-SITR had convergent validity, as they shared statistically significant positive correlations. Specifically, the K-SITR-inferred case severity was related to the participants' self-injury duration ($r$ = .29), showing that having a longer self-injury history was related to more severe self-injury than self-injury for a shorter period. The K-SITR-inferred expected negative treatment outcome

**Table 5. Correlations between K-SITR, self-injury questionnaires, and related factors.**

|  | K-SITR severity | K-SITR outcome | SI duration | SI frequency | FASM | SHI | GSI |
|---|---|---|---|---|---|---|---|
| **K-SITR outcome** | .75** | | | | | | |
| **SI duration** | .29 | .24 | | | | | |
| **SI frequency** | .22 | .38** | .22 | | | | |
| **FASM** | .03 | .16 | .26 | .34* | | | |
| **SHI** | -.09 | .05 | -.12 | .19 | .66** | | |
| **GSI** | -.06 | -.13 | -.10 | .04 | .25 | .50** | |
| **Externalization** | -.12 | -.02 | .014 | .36* | .31* | .34* | .31* |

K-SITR severity = K-SITR-inferred case severity, K-SITR outcome = K-SITR-inferred expected (negative) treatment outcome, SI = self-injury, externalization = physical aggression measured by AQ.

$* \ p < .05$

$** \ p < .01.$

was related to participants' self-injury frequency ($r = .38$), showing that engaging in self-injury more frequently was related to a greater likelihood of experiencing prolonged aftereffects than engaging in self-injury less frequently.

The SHI did not show significant correlations with the participants' self-injury duration and frequency ($r = -.12$, and .19, respectively). Instead, the SHI had a significant positive correlation with mood issues ($r = .5$), whereas the K-SITR and FASM did not. The FASM showed significant but smaller positive correlations with the duration and frequency of self-injury ($r = .26, .34$ respectively) than the K-SITR.

The K-SITR was found to have discriminant validity from the aggression externalized toward others, as they shared insignificant correlations ($r = -.12$ with the K-SITR-inferred severity, and -.02 with the treatment outcome). FASM and SHI had significantly positive correlations ($r = .31, .34$, respectively) with the likelihood of externalization.

The K-SITR had incremental validity in explaining participants' self-injury frequency compared to the FASM and SHI (Table 6). The K-SITR explained 8.3% more of the variance than the FASM (11.8%), which made the result statistically significant at an alpha level of .01. SHI alone did not explain a significant amount of the variance in self-injury frequency (3.5%); however, the inclusion of K-SITR significantly increased and succeeded in explaining 13.9% of the variance ($p < .05$).

Finally, the counseling expert offered comments on the K-SITR's topic titles in terms of their adequacy in explaining clients during counseling. After an open discussion between the authors, the three topic titles were revised with prudence (Table 7). The title of topic 3 in the

**Table 6. Results of hierarchical regression to check for K-SITR's incremental validity.**

| Step | Assessment | B | SE | β | t | F | $R^2$ | $\triangle R^2$ |
|---|---|---|---|---|---|---|---|---|
| 1 | FASM | .05 | .02 | .34 | 2.45* | 6.00* | .12 | .12 |
| 2 | FASM | .05 | .02 | .31 | 2.31* | 5.51** | .20 | .08 |
|  | K-SITR | .60 | .28 | .29 | 2.13* | | | |
| 1' | SHI | .05 | .04 | .19 | 1.27 | 1.61 | .03 | .03 |
| 2' | SHI | .05 | .04 | .19 | 1.36 | 3.57* | .14 | .11 |
|  | K-SITR | .67 | .29 | .32 | 2.32* | | | |

$* \ p < .05$

$** \ p < .01.$

**Table 7. Revised topic titles by accommodating comments from a counseling expert.**

| Sub-set | Index | Title before revision | Title after revision |
|---------|-------|----------------------|---------------------|
| A | 3 | Adolescence psychological issues and delinquency | Adolescence psychological issues |
| A | 9 | Female struggles | Peer relationship struggles |
| A | 4 | Familial conflicts 2 | Overcontrolling home environment |

training data subset A omitted "delinquency" to prevent from giving counselors negatively nuanced information about the client. The title of topic 9 in subset A was changed from "Female struggles" to "Peer relationship struggles" to not limit the peer bullying victims to females. The title of topic 4 in subset A was changed from "Familial conflicts 2" to "Overcontrolling home environment" to better reflect the topic's key tokens like "fault finding," "cellphone" (parents restricting children's cellphone use), and "calves" (a sign of physical punishment) and distinguish it from other topics about unstable home environment.

## Discussion

This study developed and validated the K-SITR, an automated machine for assessing the levels of case severity and expected treatment outcomes from the natural writings of Korean youths who self-injure. Based on a set of high-quality Internet posts on self-injury experiences, the K-SITR succeeded in emulating counseling experts' judgments about written cases. The results showed that entering a paragraph-length writing sample by a self-injuring client into the K-SITR can provide a structured report (S1 File) on the client's likely condition, along with practical suggestions such as intervention priorities.

Study 1 classified a given writing condition in comparison to training data subsets. During the development process, the subset topic models showed meaningful characteristics of Korean youths who wrote Internet posts affiliated with each subset. The topic model of subset A, "serious but positive outcome expected," showed fewer references of psychopathological symptoms compared to the cases belonging to subset B, "serious and negative outcome concerning." Subsets A and B were similar in terms of having mere familial support; however, expressions of self-stigma were highlighted in subset B. Subsets C, "not as serious and positive outcome expected," and D, "not as serious but negative outcome concerning," showed some interest in seeking professional help. Subset C consisted of descriptions of specific contexts and reasons for familial conflict, and subset D focused on expressing negative emotions such as lethargic feelings, a sense of alienation, and inadequacy.

These results are consistent with the literature on youth self-injury in terms of the risks and maintenance of self-injury [1, 20, 47, 61]. As found in the subset topic models, youths with unstable home environments are at a greater risk of engaging in self-injury (common across subsets A through D) [1, 47] but experiencing self-injury is not always related to psychopathology, as shown in Swannell et al.'s [2] meta-analysis of the nonclinical population. Instead, repeated self-injury during stressful life events can increase the risk of developing psychological symptoms (a difference between subsets A and B) and self-stigma (found only in subset B) over time [6]. Studies on youths have emphasized that self-injury intervention is challenging [19]. Because of the youths' concerns over social stigma and rejection [62], more severe self-injury cases (cases in subsets A and B) tend to conceal their identity and have less interest in seeking psychological treatment compared to less severe cases (cases in subsets C and D) [19]. These results are consistent with the literature, suggesting that online text data have representative value for research [32], under the condition that the data are retrieved and used according to the study objectives.

A few topics found in subsets C and D showed results specific to Korean culture. While risk factors such as unstable home environments and trauma shown in subsets A and B corresponded with those of Western youth [1, 47], pressure for academic success was found to be one of the most frequently cited sources of stress in Korean youths. Their academic status and career choices led to familial conflict. These results are consistent with those of previous studies on Korean youths [34, 53]. Researchers have proposed that excessive academic pressure at home with weak parental support can raise youths' stress to the level that they injure themselves in a collectivistic and highly competitive culture like South Korea (4th major stressor reported by young Koreans who self-injured [63]).

Study 1 attempted to reflect on conventional psychological steps in the development of a measure. This interdisciplinary study began by exploring a corpus of 16,645 Internet posts, such as how a typical psychological study would start from a literature review or in-depth interviews, to gain an overall understanding of the phenomenon of interest. Second, this study selected training data based on a survey of their suitability for the study's objectives, as a typical study would brainstorm and select pilot items based on prior knowledge of the construct. Similar to the two-step factor analysis with different samples in a typical study, this study attempted to refine the procedure by adding a step for beta testing and revision. Different groups of counselors participated in the first survey for the initial coding of the machine and the second survey for the beta test to gain external validity.

Study 2 validated the K-SITR to enhance its utility in counseling practices. Prior studies conducted in the computer programming and data science fields have attempted to provide psychologically meaningful machines and modules [27, 28]; however, they have rarely been accepted by psychologists because of their lack of psychological consideration and validation procedures [32]. Thus, this study confirmed the convergent, discriminant, and incremental validity of the K-SITR compared to the FASM and SHI self-injury questionnaires and other related psychological factors. In addition, a counseling expert confirmed the face validity of the K-SITR.

Specifically, the levels of case severity and expected (negative) treatment outcomes inferred by the K-SITR converged with the two most important clinical criteria—self-injury frequency and duration. The FASM also converged with the criteria but to a smaller extent. In contrast, SHI was more closely related to the symptoms of mood disorders than to self-injury itself. This result implies that the SHI items about indirect self-harm (e.g., torturing oneself with overly self-critical thoughts) may have discouraged its validity in assessing the characteristics specific to self-injury, despite its wide use in research and practice. Moreover, the K-SITR demonstrated significant incremental validity over the FASM and SHI. Although the FASM and SHI assess respondents' self-injury severity directly by asking about the number of times they have used different self-injury methods, the incremental validity of the K-SITR over self-report measures shows that direct questioning may not be essential for assessing self-injury severity. The results showed that assessing naturally written texts can provide results as meaningful as traditional questionnaires, with less risk of provoking self-concealing tendencies in self-injuring respondents.

This study has a unique value in the literature as an example of interdisciplinary research. The strategic use of online text data can provide meaningful knowledge about a population and/or phenomenon, especially when the study topic is not socially sanctioned, such as self-injury and other suicidal behaviors or delinquency [32, 63]. This study purposefully integrated computer programming with surveys of counseling experts and young Koreans to present the K-SITR in a psychologically valid manner. In addition, the K-SITR has practical value for counseling self-injuring clients. Acquiring a short writing sample from a client and testing it using the K-SITR prior to intake can provide a tentative outlook on this case. Deciding on

intervention strategies requires an in-depth exploration of the client; however, the value of the K-SITR lies in providing options, especially in settings where professionally licensed counselors are not always available. Counselors can use these results to prepare a list of important questions regarding suicidal intent or experiences of trauma. Unlike questionnaires that directly ask about clients' self-injury methods and frequencies, the K-SITR could be a useful tool to reduce clients' self-concealing tendencies.

The advanced and generalized use of the K-SITR could further enhance its efficiency. The K-SITR can assess the writings of many people as an entire school or community, and print structured reports at once 100% virtually. In addition, it can be administered multiple times longitudinally without causing fatigue in clients because of repeated measures. As the K-SITR is not bound to survey items, any natural writing about one's self-injury experience can be assessed. When applied to actual counseling, changes in a client's topic profile could provide hints to counselors regarding recovery. When updating the K-SITR in future studies, recovery-related topics could be added and used to adjust the score of the expected negative treatment outcome to elaborate on the inferential procedure of the K-SITR.

## Limitations and suggestions for future studies

This study has a few limitations that should be addressed in future studies. First, there are some limitations related to the text-mining approach. Although this study used anonymous writing that was openly shared and left online, it is ethically desirable to obtain informed consent for future studies using the K-SITR. When administering the K-SITR in schools and private services, consent should be obtained before assessment. The K-SITR was developed on the basis of 178 high-quality posts. Although studies such as that by Jain et al. [64] emphasize that the quality of training data can surpass its quantity, future studies could supplement more writings retrieved either from online or counseling clients (with informed consent) to gain generalizability.

Second, data characteristics may affect generalizability. While Study 1 shows that online text data can be a useful source for psychological research, online self-disclosure may differ somewhat from offline self-disclosure. One possibility is that anonymity allows for the exaggeration of psychological pain [65, 66]. Future studies should focus on examining the reliability of online data in direct comparison with offline survey responses. In addition, other topic modeling techniques such as the random projection method [42] could be considered as an alternative to LDA for analyzing short texts such as this data in this study. A comparative analysis of several commonly used topic modeling methods with Korean textual datasets (as done so with English datasets in [45]) would help future multidisciplinary researchers choose a method that best fits the Korean language.

Third, the K-SITR was developed based on a set of spontaneous writings found online but aimed to assess writings obtained (mainly) at the beginning of a psychological service, such as during an intake session. Although the K-SITR does not require structured writing and is, therefore, relatively free of social desirability concerns compared to self-report questionnaires, field research is needed to examine its efficacy from the perspective of actual clients and practitioners. In such studies, verbal prompts to clients who self-injure should be cautious not to provoke the clients' tendency toward self-concealment and to guide them to be as open as possible about their self-injury experiences, such as when they are online. To this end, the counselor or an intake interviewer is encouraged to inform the clients that all topics and related thoughts are welcome and, most importantly, that their writings will be used to better understand their struggles. Informing clients that the purpose of collecting the writings is not to

make a definitive diagnosis would help alleviate the clients' concerns about premature and potentially stigmatizing judgements.

Lastly, further studies are needed to improve the usefulness of the K-SITR. This study placed the greatest emphasis on presenting the K-SITR with utility for counseling practice. However, the results of the K-SITR are only preliminary and based on the first impression of a client's presenting concerns as they appear in their writing; therefore, its use is limited to the early phase of counseling when quick judgments are needed. Future studies should use a longitudinal approach to examine whether the K-SITR is generalizable to assess counseling processes and outcomes. Finally, a natural language-based machine requires updates over time. Like questionnaires requiring revisions, the K-SITR should be regularly updated to omit outdated topics/tokens (e.g., Covid-19-related tokens) and supplement them with the latest information about the self-injuring youth. At the time of revision, validation of the updated version of the K-SITR would require a larger sample size with ongoing self-injury than that used in Study 2 ($N$ = 50) to ensure the performance of the K-SITR over other self-report measures. Collaborative research with a school counseling center or hospital could help recruit young, self-injuring participants under the protection of their therapists.

## Conclusion

This study developed and validated an automated machine named the K-SITR that can assess self-injury-related information from Korean youths' natural writings about their self-injury experiences. Based on a corpus of Internet posts, the K-SITR can infer a case's severity and expected treatment outcome, as counselors can make therapeutic judgments about their clients. After revisions, the K-SITR showed no statistically significant differences from the counselors' judgments. The K-SITR was further validated using a sample of young Koreans who had experienced self-injury. The results confirmed that the K-SITR had convergent, discriminant, and incremental validity compared to popular self-injury questionnaires (FASM and SHI). This study provides a valuable example of interdisciplinary research. The K-SITR is expected to have practical utility in counseling practices.

## Supporting information

**S1 Table. K-SITR codebook example.** Key = key tokens; rep. score = tokens' representative scores; prop. est. = topic proportion estimates.
(DOCX)

**S1 File. K-SITR client report example.**
(DOCX)

## Author Contributions

**Conceptualization:** Seoyoung Kim.

**Data curation:** Seoyoung Kim.

**Formal analysis:** Seoyoung Kim.

**Investigation:** Seoyoung Kim.

**Methodology:** Seoyoung Kim.

**Project administration:** Dong-gwi Lee.

**Resources:** Dong-gwi Lee.

**Software:** Seoyoung Kim.

**Supervision:** Dong-gwi Lee.

**Validation:** Seoyoung Kim.

**Writing – original draft:** Seoyoung Kim.

**Writing – review & editing:** Dong-gwi Lee.

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
