## [Decision Letter · Decision Letter 0]

28 Aug 2024

PONE-D-24-27189Development and validation of an automated machine for self-injury assessment via young Koreans’ natural writingsPLOS ONE

Dear Dr. Lee,

Thank you for submitting your manuscript to PLOS ONE. After careful consideration, we feel that it has merit but does not fully meet PLOS ONE’s publication criteria as it currently stands. Therefore, we invite you to submit a revised version of the manuscript that addresses the points raised during the review process.

Please carefully check the Reviewers' comments and improve the manuscript. Reviews provide details into areas that require improvements. 

We look forward to receiving your revised manuscript.

Kind regards,

Agnieszka Konys, Ph.D.

Academic Editor

PLOS ONE

Journal requirements: 1. When submitting your revision, we need you to address these additional requirements. Please ensure that your manuscript meets PLOS ONE's style requirements, including those for file naming. The PLOS ONE style templates can be found at https://journals.plos.org/plosone/s/file?id=wjVg/PLOSOne_formatting_sample_main_body.pdf and https://journals.plos.org/plosone/s/file?id=ba62/PLOSOne_formatting_sample_title_authors_affiliations.pdf. 2. Your ethics statement should only appear in the Methods section of your manuscript. If your ethics statement is written in any section besides the Methods, please move it to the Methods section and delete it from any other section. Please ensure that your ethics statement is included in your manuscript, as the ethics statement entered into the online submission form will not be published alongside your manuscript. 

Reviewers' comments:

Reviewer's Responses to Questions

**Comments to the Author**

1. Is the manuscript technically sound, and do the data support the conclusions?

Reviewer #1: Yes

Reviewer #2: Yes

2. Has the statistical analysis been performed appropriately and rigorously? 

Reviewer #1: Yes

Reviewer #2: Yes

3. Have the authors made all data underlying the findings in their manuscript fully available?

Reviewer #1: Yes

Reviewer #2: No

4. Is the manuscript presented in an intelligible fashion and written in standard English?

Reviewer #1: Yes

Reviewer #2: Yes

5. Review Comments to the Author

Reviewer #1: This paper provides empirical results on the K-SITR software, developed by the authors, for assessing self-harm in individuals from their writing and forecasting the resulting outcomes. The authors developed this natural language processing software using off-the-shelf methods and in collaboration with counsellors and therapists. An analysis of K-SITR results with several questionnaire-based scoring mechanisms was provided.

Overall, the paper seems scientifically sound and could be published with minor revisions.

1. It is never made clear how the matching is actually done (e.g., on Page 11). Is it cosine similarity or some kind of common word count? Is there a projection onto topic space occurring? This should be spelled out.

2. Similar question on Page 12: It seems like a raw word count comparison was performed. There are other ways of doing this, including projection methods (onto the reduced topic space). Were they considered?

3. On Page 28, the authors suggest acquiring a writing sample on intake for patients. However, an arbitrary writing sample may not be sufficient. What prompts are suggested and what (future work) methods could be used to identify prompts that do not provoke self-censoring? Since K-SITR was developed for open-ended writing with no prompt, what changes adjustments, if any, may need to be made?

Minor Comments:

Page 7, Line 133: "Raw corpus..." should be "The raw corpus..."

Page 9, Line 181: There is a tense inconsistency. The sentence should read, "Each training dataset was subjected..."

Page 10, Line 193: "opinions" should be "topics".

Page 10, Line 199: "find more..." should be: "See Blei et al. [41] for more on LDA."

Page 10, Line 201: "...and within-topic..." should be "...and small within-topic..."

Page 11, Line 213, "of the machine..." should be "of the machine learning algorithm..."

Page 13, Line 264: "0.64 a more..." should be "0.64 higher..." or something like it.

Page 17, Line 319: "eliciting urges" is not right. Do you mean, "encouraging"?

Reviewer #2: The following issues should be considered:

1. Abstract: The abstract is too long. You should check the journal's format.

2. Keywords: Please add 5 keywords. It is advisable to avoid abbreviations in this section.

3. Introduction: This section is too long. Please rewrite the main contributions of the research at the end of the Introduction. It is better to write it more concisely and in a more substantial form. Currently, this part resembles a related work section. You should add a new section titled 'Related Work,' move the relevant text there, and then write a new Introduction.

4. Related Work: The related work section must be completed and updated with recent articles from reputable journals.

5. Methodology and Proposed Work: This section must be added. Also, the applications of this method have yet to be mentioned.

6. Section Numbering: The sections should be numbered.

7. Comparative Analysis: One of the fundamental questions is, what are the advantages and disadvantages of your method compared to recent methods?

8. Conclusion: The conclusion section is acceptable.

9. Article Formatting: In general, the format of articles may vary across different fields. Therefore, I suggest reading several articles within the current field of study and considering their format for structuring the various sections of your paper. For example, this could include aspects such as: 1. Naming the sections, 2. Ordering the sections, 3. Numbering the sections.

10. Additional Work: There are some other works on semantic analysis by LDA that should be considered:

https://doi.org/10.1155/2022/7612276

https://doi.org/10.1142/S0219622022500584

https://doi.org/10.1111/exsy.12527

6. PLOS authors have the option to publish the peer review history of their article (what does this mean?). If published, this will include your full peer review and any attached files.

Reviewer #1: **Yes: **Christopher Griffin

Reviewer #2: No

---

## [Author Response · Author response to Decision Letter 0]

28 Nov 2024

We sincerely appreciate the opportunity to revise our manuscript. 

We attached our response to each review comment along with our revised manuscript. Plus, we are submitting our response here too: 

Response to Reviewer 1

Comment 1: It is never made clear how the matching is actually done (e.g., on Page 11). Is it cosine similarity or some kind of common word count? Is there a projection onto topic space occurring? This should be spelled out.

Response 1: Thank you for your help in clarifying the matching process. We have clarified the two-step matching process of the K-SITR machine (visualized in Fig. 2 as H1 and H2) and elaborated it with examples, on pages 16-18, lines 310-360 in our revised manuscript with track changes. The H1 matching aimed to match a new text case (presumably, a new counseling client’s writing) to one of the subsets to tentatively assess the severity of the new case. To do this, the K-SITR counts the frequencies of word-to-word matches. And within the subset, the H2 matching aimed to match the new case with a specific case that the K-SITR has as training data. To do this, the K-SITR compares the topic profile of the new case with the topic profile of each training data case until it finds the most similar training data case. As a result of this matching, the K-SITR can retrieve the professional counselor judgements that were made on the subset and the best matching training data case for the new client’s case.

Comment 2: Similar question on Page 12: It seems like a raw word count comparison was performed. There are other ways of doing this, including projection methods (onto the reduced topic space). Were they considered?

Response 2: The reason for choosing the LDA method is now explained with supporting references on page 15 lines 293-297. Following your kind suggestion, we have also mentioned the random projection method for short text data as an alternative topic modeling method for future studies in our Discussion on page 35, lines 691-696. 

Comment 3: On Page 28, the authors suggest acquiring a writing sample on intake for patients. However, an arbitrary writing sample may not be sufficient. What prompts are suggested and what (future work) methods could be used to identify prompts that do not provoke self-censoring? Since K-SITR was developed for open-ended writing with no prompt, what changes adjustments, if any, may need to be made?

Response 3: Thank you for bringing up a very important point. We have added a guide for counselors on the verbal prompts that can help self-injuring clients feel as open as possible, on pages 35-36, lines 702-710.

Comment 4: Minor Comments:

Page 7, Line 133: "Raw corpus..." should be "The raw corpus..."

Page 9, Line 181: There is a tense inconsistency. The sentence should read, "Each training dataset was subjected..."

Page 10, Line 193: "opinions" should be "topics".

Page 10, Line 199: "find more..." should be: "See Blei et al. [41] for more on LDA."

Page 10, Line 201: "...and within-topic..." should be "...and small within-topic..."

Page 11, Line 213, "of the machine..." should be "of the machine learning algorithm..."

Page 13, Line 264: "0.64 a more..." should be "0.64 higher..." or something like it.

Page 17, Line 319: "eliciting urges" is not right. Do you mean, "encouraging"?

Response 4: Thanks for your kind suggestions, all sentences are now revised except one on page 15, line 298. 

We meant to say, “An efficient topic model has small between-topic similarities and large within-topic similarities,” and the previous sentence had typos. We apologize for the confusion. 

Response to Reviewer 2 

Comment 1: The abstract is too long. You should check the journal's format.

Response 1: Following your advice and PLOS ONE’s recent publications, our abstract has now been shortened and revised. 

Comment 2: Please add 5 keywords. It is advisable to avoid abbreviations in this section

Response 2: Five keywords are added after the abstract. 

Comment 3: This section is too long. Please rewrite the main contributions of the research at the end of the Introduction. It is better to write it more concisely and in a more substantial form. Currently, this part resembles a related work section. You should add a new section titled 'Related Work,' move the relevant text there, and then write a new Introduction.

Response 3: Thank you for helping us restructure the Introduction to improve readability. We have revised the entire Introduction from page 4 to page 11. Specifically, we made the Introduction more concise by omitting parts that had unclear relevance to our research topic and objectives. Following your suggestion, we added the sections “Efforts to Assess Self-Injury and Related Work” (on page 6 line 99 in our revised manuscript with track change) and “Methodology and Proposed Work” (on page 7, line 127) to distinguish the background and literature review from the part where we introduce our study and expected implications. 

Comment 4: The related work section must be completed and updated with recent articles from reputable journals.

Response 4: Following your suggestion, we added the section “Efforts to Assess Self-Injury and Related Work” (on page 6, line 99) and a new reference [22]. Also, in the following “Methodology and Proposed Work” section, we have elaborated on our methodological approach by adding research information from some of the previous studies that we have referenced (e.g., [30, 31]).

Comment 5: This section (“Methodology and Proposed Work”) must be added. Also, the applications of this method have yet to be mentioned.

Response 5: Regarding your third suggestion, we have added the sections “Efforts to Assess Self-Injury and Related Work” (on page 6, line 99) and “Methodology and Proposed Work” (on page 7, line 127) to distinguish the background and literature review from the part where we present our study (objectives and methods) and expected impact. 

Comment 6: The sections should be numbered.

Response 6: We have revised our heading format based on the PLOS ONE submission guidelines. (We found the PLOS ONE does not allow psychology articles to use numbered sections/headings. Please see: https://doi.org/10.1371/journal.pone.0290756)

Comment 7: One of the fundamental questions is, what are the advantages and disadvantages of your method compared to recent methods?

Response 7: Thank you for your help in assuring us of the benefits of taking the text-mining approach to self-injury assessment. We have added a paragraph on the methodological advantages on page 10, lines 185-198, compared to other methods including the self-report surveys and previous text-mining results that did not reflect the opinions of psychologists. Potential disadvantages/limitations are explained in the Discussion, primarily on pages 34-35, lines 678-696. 

Comment 8: The conclusion section is acceptable.

Response 8: N/A

Comment 9: In general, the format of articles may vary across different fields. Therefore, I suggest reading several articles within the current field of study and considering their format for structuring the various sections of your paper. For example, this could include aspects such as: 1. Naming the sections, 2. Ordering the sections, 3. Numbering the sections.

Reseponse 9: We have reconstructed and revised our sections/headings based on the recent PLOS ONE publications in the field of psychology (e.g., https://doi.org/10.1371/journal.pone.0290756). 

Comment 10: There are some other works on semantic analysis by LDA that should be considered:

https://doi.org/10.1155/2022/7612276

https://doi.org/10.1142/S0219622022500584

https://doi.org/10.1111/exsy.12527

Response 10: Thank you for your thoughtful suggestion.

---

## [Editor Report · Decision Letter 1]

15 Dec 2024

Development and validation of an automated machine for self-injury assessment via young Koreans’ natural writings

PONE-D-24-27189R1

Dear Dr. Lee,

We’re pleased to inform you that your manuscript has been judged scientifically suitable for publication and will be formally accepted for publication once it meets all outstanding technical requirements.

Kind regards,

Agnieszka Konys, Ph.D.

Academic Editor

PLOS ONE
---

## [Editor Report · Acceptance letter]

19 Dec 2024

PONE-D-24-27189R1 

PLOS ONE

Dear Dr. Lee, 

I'm pleased to inform you that your manuscript has been deemed suitable for publication in PLOS ONE. Congratulations! Your manuscript is now being handed over to our production team.

Kind regards, 

on behalf of

Dr. Agnieszka Konys 

Academic Editor

PLOS ONE